# Mechanistic Insights Revealed by YbtPQ in the Occluded State

**DOI:** 10.3390/biom14030322

**Published:** 2024-03-08

**Authors:** Wenxin Hu, Chance Parkinson, Hongjin Zheng

**Affiliations:** Department of Biochemistry and Molecular Genetics, School of Medicine, University of Colorado Anschutz Medical Campus, Mail Stop 8101, Aurora, CO 80045, USA

**Keywords:** cryo-EM, yersiniabactin importer YbtPQ, occluded, ABC transporter

## Abstract

Recently, several ATP-binding cassette (ABC) importers have been found to adopt the typical fold of type IV ABC exporters. Presumably, these importers would function under the transport scheme of “alternating access” like those exporters, cycling through inward-open, occluded, and outward-open conformations. Understanding how the exporter-like importers move substrates in the opposite direction requires structural studies on all the major conformations. To shed light on this, here we report the structure of yersiniabactin importer YbtPQ from uropathogenic *Escherichia coli* in the occluded conformation trapped by ADP-vanadate (ADP-Vi) at a 3.1 Å resolution determined by cryo-electron microscopy. The structure shows unusual local rearrangements in multiple helices and loops in its transmembrane domains (TMDs). In addition, the dimerization of the nucleotide-binding domains (NBDs) promoted by the vanadate trapping is highlighted by the “screwdriver” action at one of the two hinge points. These structural observations are rare and thus provide valuable information to understand the structural plasticity of the exporter-like ABC importers.

## 1. Introduction

ABC transporters, despite comprehensive research, continue to be a subject of ongoing discovery. Recent findings indicate that three bacterial importers exhibit the characteristic structural fold associated with type IV ABC exporters [1]. This observation represents a departure from the previously understood configurations of ABC importers. These importers are yersiniabactin (Ybt) importer YbtPQ from uropathogenic *Escherichia coli* (UPEC) [2] as well as IrtAB and Rv1819c from *Mycobacterium tuberculosis* [3,4]. Such studies have raised immediate interest in understanding how these importers function similarly/differently compared with the known type IV ABC exporters. To carry this out, structural studies of these exporter-like importers in all their major physiological conformations are necessary. Here, we have continued our study on YbtPQ and expect to shed more light on the topic.

Initially discovered in *Yersinia enterocolitica* [5], Ybt is a virulence-associated siderophore molecule synthesized in many pathogens such as UPEC. UPEC is the leading cause of urinal tract infections (UTIs) that affect about 150 million people worldwide each year [6,7]. During the infections, UPEC secretes multiple siderophores including Ybt. Ybt can recognize and chelate various physiologically relevant metal ions such as Fe^3+^, Cu^2+^, Co^2+^, Ni^2+^, Ga^3+^, and Cr^3+^ [8]. The metal-chelated Ybt is then imported back into the bacteria through the outer membrane importer FyuA [9] and inner membrane ABC importer YbtPQ [10]. This process of siderophore-mediated metal uptake is one of the most prevalent and vital biological processes in microbes [11,12]. Specifically, in UPEC, deleting genes encoding the YbtPQ importer has a detrimental effect on the Ybt-mediated metal uptake and later leads to profoundly decreased UTIs in a cystitis mouse model [13]. Previously, we determined the high-resolution structure of YbtPQ in the inward-open conformation in both the apo (PDB:6P6I) and substrate-bound (PDB: 6P6J) states. The structures have not only revealed its exporter-like fold but also provided initial information about the substrate binding and releasing [2].

To better understand the dynamics of YbtPQ, it is necessary to reconstruct its entire transport cycle. In this study, we successfully trapped YbtPQ in the occluded state using ADP-Vi in the lipid nanodisc environment. We then determined the structure of the occluded YbtPQ to the 3.1 Å resolution using cryo-electron microscopy single particle reconstruction. In the TMDs, the structure shows a closed periplasmic gate of the translocation pathway similar to that in the inward-open conformation. However, there are considerable conformational changes in at least five transmembrane helices (TMs) and two loops of the TMDs. In the meantime, the cytosolic gate of the pathway is closed by three groups of residues through strong interactions. The NBDs in the occluded state containing sharp experimental densities for the two ADP-Vi molecules are further dimerized compared with those in the inward-open state. Interestingly, this further dimerization is achieved by the ~180° rotation action of the C-terminal YbtQ helix coupled with lateral movement, thus acting like a screwdriver. As far as the authors know, such a “screwdriver” mode of NBD dimerization has never been reported for any ABC transporters.

## 2. Results and Discussion

### 2.1. Structure Determination of YbtPQ in the Occluded Conformation

Here, we expected to trap YbtPQ in unknown conformations using orthovanadate as it had been demonstrated to be an effective ATP analog to trap various ATPases, such as myosin [14], and dynein [15], as well as multiple ABC transporters [16,17,18]. To carry this out, we used YbtPQ incorporated in the lipid nanodiscs with *E. coli* polar extract lipids (YbtPQ-nanodisc) because this sample was shown to be functional in our previous study [2]. After vanadate trapping, the ATPase activity of the YbtPQ-nanodisc dropped by ~90% (Appendix A). Then, we used this sample for single particle cryo-EM analysis and obtained a final reconstruction at a 3.1 Å resolution (Appendix A and Figure 1A, Appendix A). The map has a size of 120 Å × 75 Å × 60 Å, which is smaller compared with that of the inward-open YbtPQ structures with a size of 120 Å × 90 Å × 60 Å. The smaller size reflects the strong dimerization between the NBDs, which also makes the cytosolic opening disappear (Figure 1B). There is no extra density in the substrate-binding pocket, thus suggesting no substrate bound. We analyzed the YbtPQ structure with the web tool MoleOnline [19]. The results show that the translocation pathway in YbtPQ is completely closed on both ends, and the diameter of the central cavity around the substrate-binding pocket is ~6 Å. Considering that the size of a Ybt-Fe^3+^ substrate is ~8 Å × 11 Å × 12 Å [20], the cavity is too small to fit any substrate. Thus, this map represents the occluded conformation of YbtPQ, which is a high-energy intermediate state after ATP binding.

### 2.2. Structural Rearrangements of Helices and Loops in the TMDs

Generally, ABC transporters with the type IV exporter fold function through the alternating access mechanism, in which the substrate-binding pocket is alternatively exposed in different conformations that are characterized by the rigid body movement of the TMDs [21,22]. In YbtPQ, TMD1 is formed by four TMs from YbtP (TM1~3, TM6) and 2 TMs from YbtQ (TM4, TM5), while TMD2 is formed by four TMs from YbtQ (TM1~3, TM6) and 2 TMs from YbtP (TM4, TM5), a typical swap of helices (subunit intertwining) in type IV ABC exporters. To reveal the structural differences in the TMDs, we carefully compared the occluded and inward-open YbtPQ structures. Here, for the inward-open conformation, although we used the structure of YbtPQ determined in detergent (6P6I), no detergent artifact was expected because the TMDs in detergent were the same as those in the lipid nanodisc, as shown in our previous publication [2]. The comparison shows that the tip positions of all 12 TMs are the same in both conformations on the periplasmic side (Figure 2A). The periplasmic gate is closed by strong hydrogen bonding among three residues from TM6 as follows: YbtQ-R296, YbtP-E297, and the backbone of YbtP-L289 (Figure 2A). Interestingly, the TMDs in YbtPQ do not simply act as two rigid bodies when switching between different conformations as there are obvious local conformational rearrangements in the TMs and related loops (Figure 2, Appendix A). Specifically, the middle of helices YbtP-TM6, YbtQ-TM6, and YbtQ-TM4-5 are completely twisted, with both ends being well aligned. The N-terminal part of YbtQ-TM3 connected to intracellular loop 1 (coupling helix 1 between the TMDs and NBDs, ICL1) moves significantly with part of the helix relaxed. In addition, the loops connecting TM5 and ICL2 (coupling helix 2 between the TMDs and NBDs) in both YbtP and YbtQ are apparently rearranged.

In our opinion, it is reasonable to postulate that the structural rearrangements happening at the end of some helices and loops are necessary to provide flexibility for any conformational changes in a protein. However, it is rather difficult to explain why so many helices in YbtPQ only twist their middle parts. In the literature, one of the closest structural homologs to YbtPQ with different available conformations is that of the heterodimeric multidrug exporter TmrAB from *Thermus thermophilus* [18,23]. In TmrAB, during conformational changes between the inward-open and occluded states, only one TM, which is TmrA-TM3, twists its middle part, while the other TMs either remain the same or only tilt one end (Appendix A). It appears that the TMs in YbtPQ are generally much more flexible compared with those in TmrAB. Thus, in YbtPQ, the flexible TMs could act like springs, making it harder to transmit all the force provided by nucleotide binding/hydrolysis in the NBDs to the periplasmic side of the TMDs. Meanwhile, the rigid nature of the TMs in TmrAB makes it easier to transmit the force. As a consequence, when trapped by ADP-Vi, the conformational switch in YbtPQ is not complete, so it only exists in an occluded state. No particles in the outward-open conformation could be found in the 2D classes (Appendix A). Meanwhile, the conformational switch in TmrAB is complete as it exists in both the occluded and outward-open states [18].

What is the mechanism by which YbtPQ closes the cytoplasmic side of the translocation pathway in its occluded state? There are three groups of residues responsible for that, all of which are around the circled area (Figure 1B). H196 and E200 from YbtP-TM4 specifically interact with K125 from YbtP-TM3 and R318 from YbtP-TM6, respectively (Figure 3A). S207 and D209 from YbtQ-TM4 interact with G125 and R123 from YbtQ-TM3, respectively (Figure 3B). Q211 from YbtP-TM4 forms strong hydrogen bonding with Q215 from YbtQ-TM4 (Figure 3C). Together, these residues help YbtPQ to remain in the occluded state.

### 2.3. Enhanced Dimerization of the NBDs by ADP-Vi

What is the nature of the interaction between ADP-Vi and the NBDs in YbtPQ? To answer this, we carefully examined the cryo-EM reconstruction and found two clear densities in the ATP-binding sites formed between the two NBDs (Appendix A and Figure 4A), which are presumably the ADP-Vi molecules. It is worth noting that the ATP and ADP-Vi densities at this resolution are indistinguishable and are structurally equivalent. The ADP-Vi molecules form canonical interactions with important residues from highly conserved loops and motifs. In specific, adenines from the two ADPs are oriented by the aromatic YbtQ-Y353 and YbtP-Y351 from the A-loops through strong π-π interactions. The ribose groups form hydrogen bonds with YbtP-Q485 and YbtQ-E486 from C-loops (ABC signature motifs). The phosphate groups interact with multiple residues from P-loops (Walker A) and C-loops. All these interactions effectively bring the two NBD domains together, so the Q-loops and H-loops are close enough to catalyze the ATP hydrolysis. Here, the orthovanadate molecules, which mimic the hydrolyzed phosphate groups, are specifically stabilized in the binding sites by residues YbtQ-Q426/YbtP-Q425 from Q-loops, YbtQ-H538/YbtP-H537 from H-loops, and YbtQ-E507/YbtP-E506, as well as the backbones of YbtP-F510/YbtQ-L512 from D-loops.

In the inward-open YbtPQ, we have shown that the NBDs are already dimerized through two sets of helix–helix interactions: the C-terminal helix of YbtP interacts with helix I514-M527 in YbtQ, and the C-terminal helix of YbtQ interacts with helix E514-L527 in YbtP (Figure 4B) [2]. It is well known that ATP (ADP-Vi in this case) binding promotes the further dimerization of the NBDs as demonstrated in many ABC transporters. This action involves one of the two known types of NBD motion as follows: the rocking motion with a pivot at the bottom of the NBDs and the swing-arm motion with a pivot in the TMDs [24]. To understand the NBD motion in YbtPQ, we extracted NBDs (residues 325–583 in YbtP and residues 326–579 in YbtQ) from both the inward-open and occluded YbtPQ structures, and we superimposed and morphed them in the UCSF chimera [20]; then, we examined the helix–helix dimerization sites. The result shows that when ADP-Vi binds, the relative positions of the C-terminal helix of YbtP and helix I514-M527 in YbtQ do not change too much. The interactions largely remain in between the last two visible turns of the C-terminal helix of YbtP and the first two turns of helix I514-M527 in YbtQ, with slight adjustments of the side chains (Figure 4B, Appendix A). Interestingly, the interactions between the C-terminal helix of YbtQ and helix E514-L527 in YbtP are drastically modified. The C-terminal helix rotates ~180° and moves along the other helix as if it is a “screwdriver” (Figure 4B, Appendix A). To date, the observed asymmetrical and screwdriver-like motion, characterized by the rotation around one helical axis and the concurrent translation along another between the two interacting helices, is unprecedented in the study of ABC transporters.

### 2.4. Concluding Remarks

In this study, we have determined the structure of the exporter-like importer YbtPQ in the occluded state with ADP-Vi being bound. The critical residues responsible for the closure of both ends of the substate translocation pathway have been discussed. By comparing the occluded structure to the inward-open structures, we have identified several unique features in YbtPQ. First, TMs in the TMDs are more flexible than other ABC transporters with a similar type IV fold as many of them twist the middle part during the conformational change. Second, NBD dimerization promoted by the vanadate trapping follows a general pivotal rocking motion. However, the two pivot points do not move symmetrically, with one going through the screwdriver-like action. 

In what manner does the newly identified structure integrate into the entire import cycle? The cycle can be analyzed in six distinct phases (Figure 5). First, we can start from the inward-open conformation without the substrate/ATP being bound, whose structure has been determined previously (PDB: 6P6I). Secondly, considering the high ATP concentration in a living cell, it is reasonable to suggest that ATP binds to YbtPQ to initiate the substrate import. Our new structure is likely to mimic this state. It seems that ATP binding could only change the protein from being inward-open to occluded, but not all the way to being outward-open. The reason for this is probably because TMs in YbtPQ are not rigid enough to transmit all the mechanical force from the NBDs up to its periplasmic gate. It is also possible that the scaffold protein restrains the lipids in nanodiscs or that the types of lipid molecules in nanodiscs are not ideal, so the periplasmic side fails to open. Another possibility is that the outward-open conformation was induced but short-lived without the presence of substrate Ybt-Fe^3+^. To test this, we performed the vanadate trapping experiment with Ybt-Fe^3+^ and analyzed the sample by cryo-EM. Unfortunately, we were not able to obtain structures other than that of the occluded state reported here. Last, the possibility that a periplasmic soluble protein may help YbtPQ switch to an outward-open state cannot be completely ruled out, although previous efforts of looking for such a partner have not been successful [8,10,25]. At this step, ATP hydrolysis brings YbtPQ back to the inward-open state, which is the basal ATPase activity (without substrate translocation) that is observed for almost all ABC transporters. Third, for reasons not fully understood yet, YbtPQ is able to reach the outward-open conformation. Fourth, the substrates are then recognized and are bound to the substrate-binding pocket exposed to the periplasm. Fifth, YbtPQ changes back to the occluded state with substrate being bound, which is most likely powered by ATP hydrolysis. Sixth, after ATP hydrolysis and with ADP being released from the NBDs, YbtPQ returns to the inward-open conformation with the substate still being bound (PDB: 6P6J). Previously, we proposed that the substrate release should be facilitated by unwinding YbtP-TM4, thereby going directly from step 6 to step 1 [2]. We would like to update this model by adding step 2 to the substrate-releasing process. After import (step 6), YbtPQ binds ATP again to promote conformational change to the occluded state (step 2). This conformational change closes the cytosolic gate, crashes the substrate-binding pocket, and is likely to provide energy to unwind helix YbtP-TM4 (also observed in the occluded structure in Figure 2B) to release the substrate. Then, YbtPQ can release ADP and switch back to step 1.

### 2.5. Experimental Procedures

Protein expression and purification. The wild-type, full-length YbtPQ importer from uropathogenic *E. coli* UTI89 was over-expressed and purified, as previously reported [2]. Briefly, the heterodimeric YbtPQ with a N-terminal His-tag on YbtP was expressed in the *E. coli* strain of C43(DE3) (Sigma-Aldrich) at 18 °C overnight with 0.2 mM Isopropyl opropyl β-D-1-thiogalactopyranoside (IPTG, UBPBio). Membrane fractions from 3 L of harvested cells were prepared in 20 mM Tris at pH 7.5, 150 mM NaCl (Buffer A), and 1 mM phenylmethylsulfonyl fluoride (PMSF). Then, 1% lauryl maltose neopentyl Glycol (LMNG) was used to solubilize YbtPQ from the membrane. YbtPQ was purified by the Ni-NTA affinity column followed by gel filtration on a Superose 6 column (GE Healthcare Life Sciences) in Buffer A with 0.002% LMNG and then concentrated to ~6 mg/mL.

Vanadate trapping of YbtPQ in lipid nanodiscs. Purified YbtPQ was reconstituted into the *E. coli* polar extract lipids (Avanti Polar Lipids) with the membrane scaffold protein MSP1D1 (Addgene), as previously reported [2]. Briefly, MSP1D1 was expressed at 37 °C in *E. coli* C43(DE3) strain, purified by one-step affinity chromatography using Ni-NTA resin, and concentrated to ~5 mg/mL. The stock of *E. coli* polar extract lipids at 50 mM was made in Buffer A with 150 mM LMNG. To reconstitute the nanodiscs, 0.5 mg purified YbtPQ, MSP1D1, and *E. coli* polar extract lipids were mixed at a molar ratio of 1:8:400 on ice for 30 min before adding 0.2 g Bio-Beads for overnight incubation at 4 °C. The assembled YbtPQ-nanodiscs were further purified by gel filtration on a Superose 6 column in Buffer A. The corresponding peak was pooled together and concentrated to ~6 mg/mL. The stock of 100 mM orthovanadate solution was prepared as previously reported [14]. Purified YbtPQ-nanodiscs were incubated with 20 mM vanadate, 18 mM ATP, and 54 mM MgCl_2_ for 15 min at 37 °C. Then, 36 mM fresh ATP was added to the solution and sat at room temperature for 5 min. The sample was finally purified by gel filtration on a Superose 6 column in Buffer A.

ATPase activity assay. The ATPase activity was determined as previously reported [2] using an ATPase/GTPase Activity Assay Kit (Sigma-Aldrich) featuring a detectable fluorescent product from malachite green reacting with the released phosphate group. The ATPase activity of YbtPQ-nanodisc trapped with various vanadate concentrations (0, 1, 2, 5, 10, and 20 mM) at 37 °C for 15 min was measured, and the graph was prepared in Excel. The experiments were repeated three times independently. 

Cryo-EM sample preparation and data acquisition. A total of 3 uL of vanadate-trapped YbtPQ-nanodiscs at ~2 mg/mL was applied to plasma-cleaned C-flat holy carbon grids (1.2/1.3, 400 mesh, Electron Microscopy Sciences) and prepared using a Vitrobot Mark IV (Thermo Fisher Scientific) with the environmental chamber set at 100% humidity and 4 °C. The grids were blotted for 3.5~4.5 s and then flash-frozen in liquid ethane, which was cooled by liquid nitrogen. Cryo-EM data was collected on a Titan Krios (Thermo Fisher Scientific) that was operated at 300 keV and equipped with a K3 direct detector (Gatan). A total of 5152 movies were recorded with a calibrated pixel size of 0.83 Å, defocus range of −1~−2.5 μm, and 50 frames with a total dose of ~60 electrons/Å^2^. 

Cryo-EM data processing, model building, refinement, and validation. The data were processed using the software Suit cryoSPARC v2.15.0 [26]. Briefly, the movies were motion-corrected using Patch motion correction, and their contrast transfer function (CTF) parameters were estimated using Patch CTF. Micrographs with a calculated defocus range of −0.8~2.8 μm, a fitted resolution of more than 8 Å, and a global motion of fewer than 35 pixels were selected for further processing. A total of ~2 million particles were picked using a Blob picker with the minimum diameter set to 60 Å and the maximum particle diameter set to 150 Å. These particles were extracted with a box size of 320 × 320 pixels and subjected to two rounds of 2D classification. A total of ~900,000 particles were selected for the ab initio reconstruction of 4~6 initial models. These initial models were heterogeneously refined, and the best group with 340,080 particles was selected for further homogeneous refinement. The final reconstruction was estimated to be at a 3.07 Å resolution based on the gold-standard [27] Fourier shell correlation (FSC) with a cut-off of 0.143. Local resolution estimation was done in cryoSPARC. The model building was carried out in the Coot program [28], which was guided by the previously published structure of YbtPQ in the inward-open conformation (PDB: 6P6J). The final rounds of model refinement were carried out by real-space refinement in PHENIX [29] with secondary structure restraints imposed. The quality of the model was assessed using MolProbity [29]. To validate the refinement, the model was refined against half-maps, and FSC curves were calculated. The final statistics are shown in Appendix A. The UCSF chimera was used to measure the map size, calculate hydrogen bonds, align and morph corresponding structural models, and prepare all figures [20].

## Figures and Tables

**Figure 1 biomolecules-14-00322-f001:**
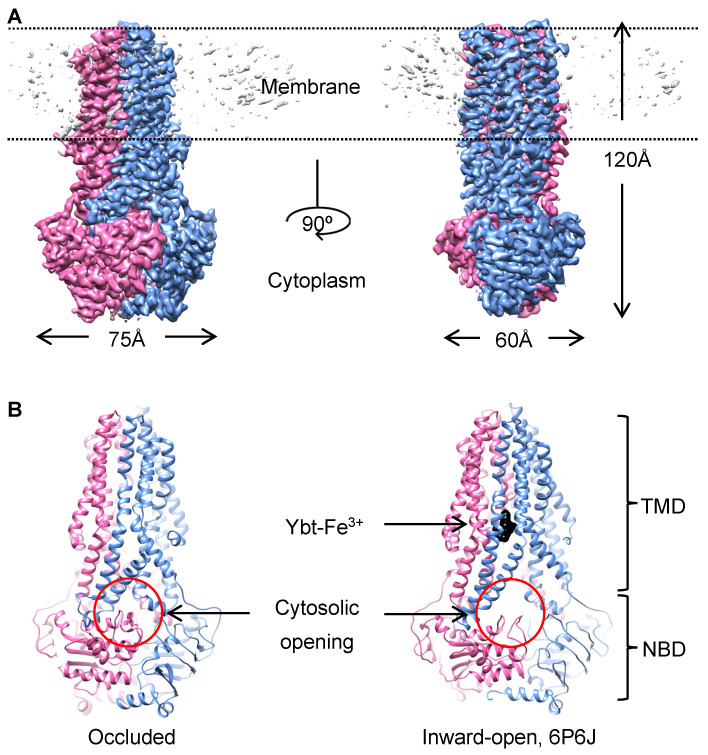
YbtPQ in the occluded state. (**A**) YbtPQ is reconstituted into the nanodiscs made of *E. coli* polar extract lipids and trapped by vanadate salt. The map of YbtPQ reconstruction at a 3.1 Å resolution is shown. YbtP is in blue, YbtQ is in pink, and disordered lipids in the nanodisc are in gray. (**B**) The cytosolic opening (red circle) to the substate-binding pocket in the inward-open YbtPQ (**right**, 6P6J) is now closed in the occluded state (**left**). The surface representation of the substrate Ybt-Fe^3+^ is shown in black.

**Figure 2 biomolecules-14-00322-f002:**
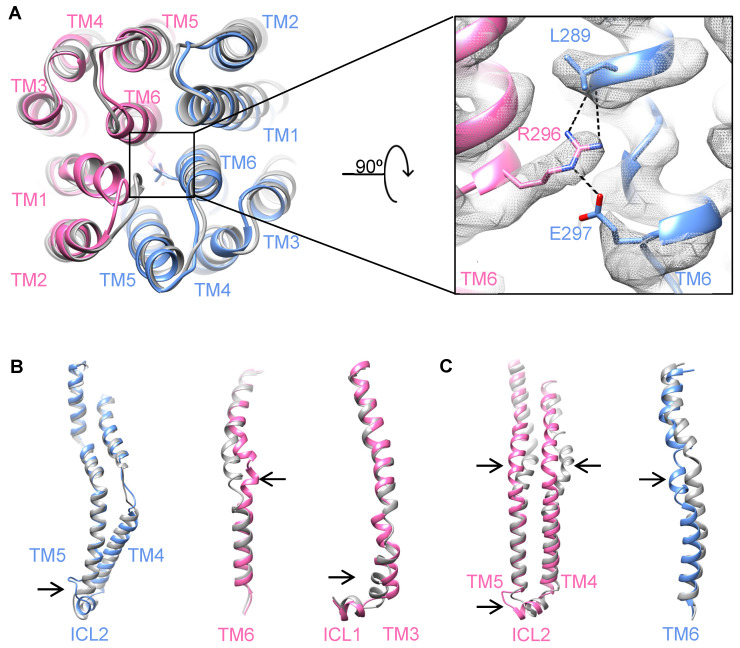
Structural rearrangement of TMDs in YbtPQ. (**A**) The periplasmic side of individual TMs in the TMDs remains almost identical in both the occluded (colored as before) and inward-open (gray, 6P6I) conformations. Residues responsible for the periplasmic gate are highlighted with their experimental densities. (**B**,**C**) TMs from TMD1 (**B**) and TMD2 (**C**) with apparent conformational changes (indicated by the arrows) are shown here.

**Figure 3 biomolecules-14-00322-f003:**
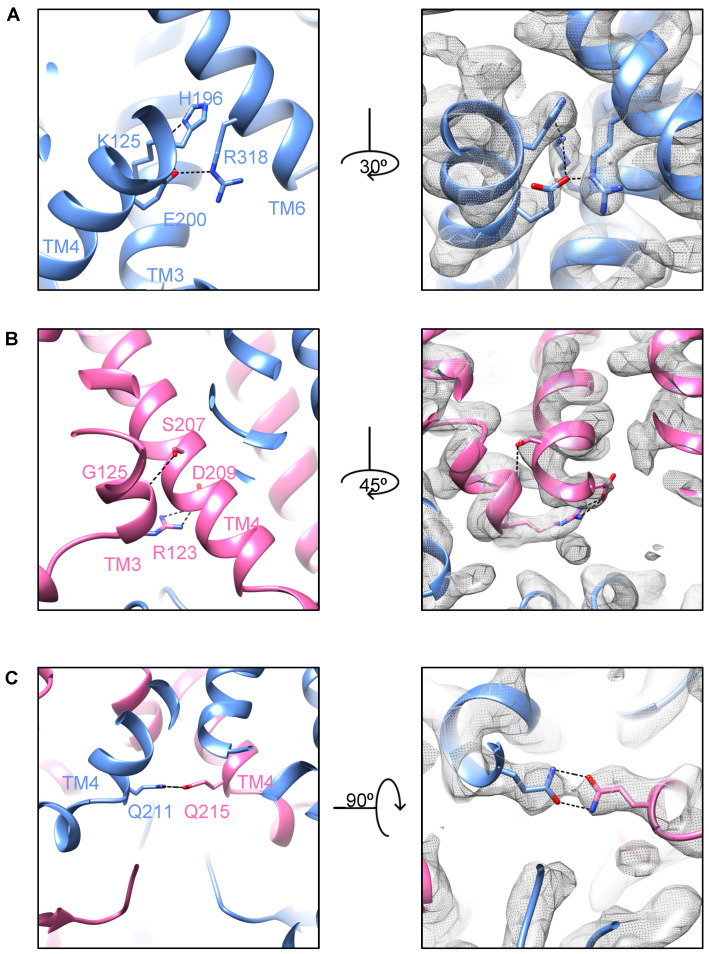
The closing of the cytoplasmic opening is mediated by three groups of residues. (**A**) Residues at the cytoplasmic side of the substrate-binding pocket from YbtP. (**B**) Residues at the back of the opening from YbtQ. (**C**) Residues directly bridge TM4s from YbtP and YbtQ.

**Figure 4 biomolecules-14-00322-f004:**
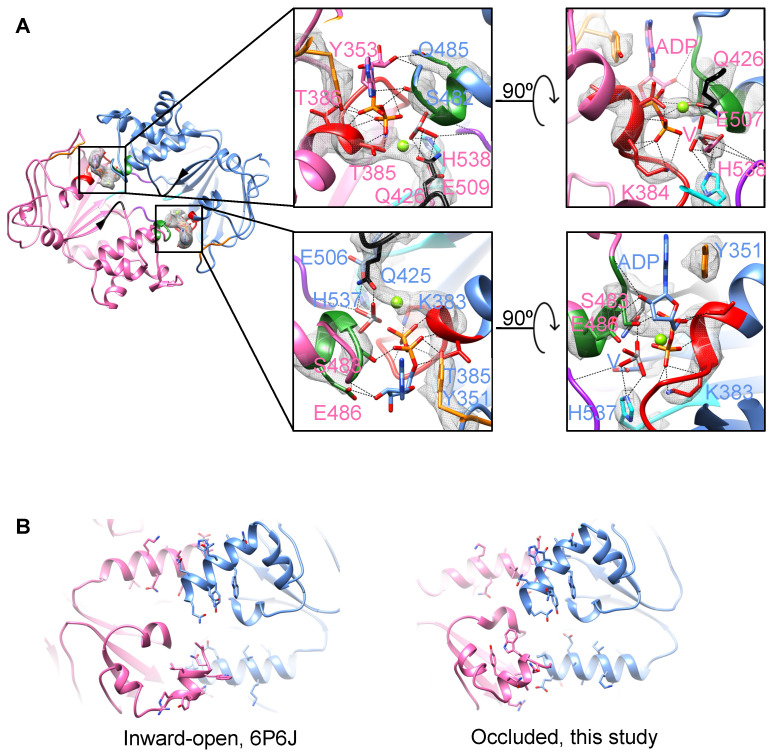
NBD dimerization upon vanadate trapping. (**A**) Detailed interactions between the ADP-Vi molecules and NBDs. Looking down the translocation pathway from the periplasm, the sliced view shows only the NBDs. The conserved P-loop, C-loop, D-loop, A-loop, Q-loop, and H-loop are colored red, green, purple, orange, black, and cyan, respectively. The experimental densities of corresponding side chains and ADP-Vi are shown as a gray mesh. (**B**) Unsymmetrical dimerization is mediated by the interactions between the C-terminal helix of an NBD and another helix of the opposing NBD. The views shown are the bottom of NBDs in the inward-open and occluded states. Side chains of the residues in those helices are shown in stick mode.

**Figure 5 biomolecules-14-00322-f005:**
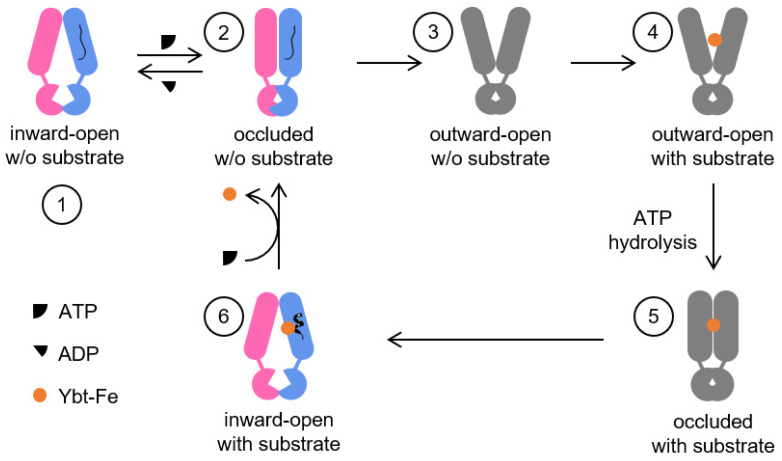
Known YbtPQ structures fit into the proposed import cycle. As discussed in the text, six conformational states of YbtPQ are indicated by the numbers. Colored cartoons represent structures determined in this and previous studies, while gray cartoons represent unknown states. The black line and coil represent YbtP-TM4 in unwinding and helical states, respectively.

## Data Availability

The cryo-EM map of YbtPQ was deposited in the Electron Microscopy Data Bank under the accession code EMD-24234. The coordinates of the atomic model of YbtPQ were deposited in the Protein Data Bank under the accession code 8VSI. All other data are available from the corresponding author upon reasonable request.

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
