# Peer review of "Mechanistic Insights Revealed by YbtPQ in the Occluded State"

_biomolecules, 2024, doi:10.3390/biom14030322_

Round 1
Reviewer 1 Report
Comments and Suggestions for Authors
Hu et al. determined the cryo-EM structure of YbtPQ in the occluded state, expanding upon the knowledge from their previous results. The results of this study were explained in a structure-centric manner, making it an ordinary structure paper with no scientific problems that explain function based on observed structure. The authors observed new structural changes that had not previously been observed in other ABC transporters and believe that these structural results will contribute to expanding knowledge of structural research in related fields. Meanwhile, some sentences in the manuscript contain numerous subjective expressions, which hinder the readability for readers. Additionally, the entire manuscript was submitted without thorough review for typos.
Major
1. The sentences below significantly reduce readability. I believe that utilizing more objective expressions will enhance readability.
- L26-27: Although ABC transporters have been extensively studied, new knowledge about these proteins is gained every day. Recently, to everyone’s surprise,
- L140-141: How does YbtPQ close the cytoplasmic side of the translocation pathway in the occluded state?
- L152-153: "How do ADP-Vi molecules interact with the NBDs in YbtPQ? To answer that, we carefully"
- L205-206 "How does the new structure fit into the whole import cycle? Let’s discuss the cycle in the following six steps (Figure 5)."
2. L191-194: "To our knowledge, this type of unsymmetrical and screwdriver-like action, rotating around the helical axis coupled with moving along another helical axis between two interacting helices, has never been reported in any other ABC transporters."
If this is the first report, I expect that the author's speculation about the biological significance of this part will provide insight into the related field.
Minor
1. The term "Title" should be excluded from the title.
2. Keywords should be included.
3. L46-47: "Previously, we have determined the high-resolution structure of YbtPQ in the inward-open conformation in both the apo and substrate-bound states." Reference and additional structural description should be added.
4. L65-67: To better understand the dynamics of YbtPQ, it is necessary to reconstruct its entire transport cycle. Previously published YbtPQ structures with (PDB: 6P6J) and without (PDB:6P6I) substrate Ybt-Fe3+ are both in the inward-open conformation 2. It is appropriate to go to the introduction.
5. Figure 4a: 90 degree angle display needs to be corrected.
6. The sentence below should be moved to method.
-L79-80: "bound. We analyzed the YbtPQ structure with the web tool MoleOnline running in the Pore Mode with default settings 19.
-L83: (calculated in UCSF chimera 20)
7. In Method: Spaces and notation errors between numbers and units must be corrected.
8. The reference format is different from the journal MDPI journal format.
Comments on the Quality of English Language
I am not a native English speaker, so I am excluding the evaluation.
Author Response
please see attached PDF file

Reviewer 2 Report
Comments and Suggestions for Authors
YbtPQ is an ABC transporter that imports yersiniabactin in complex with Fe3+ ions. However, its fold is typical of ABC exporters, which function along a mechanism of ‘alternating access’. The goal of the authors is to understand the mechanism of exporter-like ABC importers. They previously determined structures of YbtPQ in inward open conformation in the substrate-bound and apo state. Here they solved a structure of the complex trapped in an occluded conformation without substrate, obtained by using ADP-vanadate.
Structural comparisons with earlier structures show that several transmembrane helices have undergone some local rearrangements. That these TM helices are flexible makes it harder to understand how they can transmit mechanical force from ATP binding or hydrolysis. The authors speculate that an incomplete conformational switch may be the reason for the occluded state.
The authors have an interesting new structure. Whether it is part of the transport cycle remains unknown. I am not fully convinced by their proposed import cycle, and I believe it may be premature to try and draw such a cycle. Additional structures - if other conformational states are possible to isolate - and MD simulations might help in solving the mechanism, but they are beyond the scope of this study.
Comments
1. The cavity of this form is too small to accommodate yersiniabactin-Fe3+. However, this structure is supposed to represent an intermediate step in transport (high-energy state after ATP binding). Have the authors tried (and presumably failed) to obtain a structure of the complex with ADP vanadate and yersiniabactin-Fe3+?
2. Fig 4B. Supplementary figures and movies were not included in the suppl materials that could be downloaded. This is in particular important regarding the screwdriver motion described by the authors. This movement of the C-terminal helix (which appears to be a new feature for ABC transporters) is somewhat difficult to figure out from this figure alone, possibly because several residues of the C terminal helix are missing from the structure (see pbd validation report).
3. Final scheme (fig 5). Several concerns exist on this model. They are mostly related to the place of their new structure in the cycle.
ADP-vanadate is considered a mimic of the transition state of ATP hydrolysis, which in the cycle leads to an occluded state with substrate (form 5). Unfortunately, no structure of the occluded state with the substrate is available.
The authors argue that the conformational switch is incomplete due to the TMH not being rigid enough, and they propose in the text another partner in the cycle to trigger the shift to the outward open state. No such periplasmic partner has been found in spite of extensive search.
The authors further argue that this occluded state w/o substrate prevents unwanted export. This is not consistent with their cycle model, in which form #2 has no substrate. For this argument to be valid, one needs to introduce additional equilibria in the cycle.
4. Unwinding of TMH4 of YbtP was observed in the inward-open structure w/o substrate and w/o ATP. Therefore, it does not appear that helix unwinding is due to ATP binding. Is it not a consequence of rearrangements because the binding site is empty?
5. In the text (but not shown in the figure) the authors place ATP hydrolysis between states 4 and 5, and ADP release between states 5 and 6 (stated in the text but not shown in the figure). If there are evidences for these events taking place at these particular steps for this type of exporter-like importers this should be referred to in the text.
Minor points
It is not clear from fig 2 how the two TMDs are delineated.
In figure 4, residue numbers do not match those in the text.
Line 181. The term ‘bottom’ of the NBDs is not clear. I suppose the authors mean membrane distal?
Line209. ...’try to start the import event’ is a strange way of putting things. Please rephrase. Same request for …’the periplasmic side refuses to open’ … (line 215).
Author Response
please see attached PDF file

Reviewer 3 Report
Comments and Suggestions for Authors
In this work, Hu et al. solved by cryo-EM the structure of a yersiniabactin importer YbtPQ in an occluded conformation upon vanadate-induced ADP trapping. By comparing this conformation to a previously published inward-facing state, the authors gained a number of mechanistic insights, including the identification of residues putatively involved in the cytosolic and periplamic gates, and a screwdriver mechanism in the C-terminal part of the nucleotide-binding domains. An important aspect of the work is that YbtPQ is an ABC importer that adopts an exporter fold. Although biochemical work is absent to validate the findings, this new conformation will be useful to understand the transport cycle of such peculiar ABC transporters. To improve the quality of the manuscript, I have a number of comments to be addressed :
- 1) There is no cryo-EM processing workflow, this is highly unusual and it should be included as supplementary figure
- 2) Although the authors claim to see two « clear densities of ADP-Vi », can they exclude an alternative scenario in which one NBD is occupied by ADP-Vi and the other NBD is ATP-bound ? Please discuss
- 3) In Figure 4A, the numbering of all the amino acids is incomplete (exemple : E48 instead of E486 etc.). Please correct
- 4) It appears that ATP binding could only change the protein from inward-open to occluded, but not to outward-open. Besides the hypotheses of the authors, one important and in my opinion likely possibility is that the OF opening was too transient to be captured. I could envision that Ybt-Fe binding stabilizes this state. Did the authors tried to incubate the substrate before performing vanadate trapping ? Regarding the cycle in figure 5 : state 3 may be short-lived as compared to state 2, but substrate binding could stabilizes state 4 and stimulates ATPase activity. These aspects should be discussed by the authors
Minor :
- 5) Keywords are missing
- 6) The screwdriver mechanism appears quite unique, and one might expect that cross-linking in this region would be highly detrimental for the transporter functionality. This would be in sharp contrast to was observed for the type IV ABC transporter BmrA (Di Cesare et al 2024). Would it be worth discussing ?
- 7) In experimental procedures, the strain used for expression is either BL21(DE3) or C43(DE3), please correct
- 8) In experimental procedures, 0.2 mg of Biobeads is probably not sufficient for proper reconstitution, and the authors did not specify the amount of transporter used for reconstitution.
Author Response
please see attached PDF file
